# One size fits all? Transferring social mindfulness measures to HRI

**Dennis Nientimp**[1]*, **Barbara C. N. Müller**[2], **Sari R. R. Nijssen**[3], **Evelien Heijselaar**[2]

**1** Sociology Department, Groningen University, Groningen, The Netherlands, **2** Behavioral Science Institute, Communication and Media group, Radboud University, Nijmegen, The Netherlands, **3** Urban and Environmental Psychology Group, Faculty of Psychology, University of Vienna, Vienna, Austria

* d.nientimp@rug.nl

## Abstract

Applying psychological measures to Human Robot Interaction (HRI) has become increasingly common. Among these, the Social Mindfulness Paradigm (SoMi)has been used to study social mindfulness towards robots through online experiments using vignettes. This line of work indicated that humans do not show prosocial behavior towards robots. However, these findings are potentially confounded in two ways: items in the SoMi task were based on human-human interactions (HHI), not HRI, and experiments did not involve real-life interactions with robots. Addressing these methodological shortcomings, the current studies investigated whether the SoMi task is a valid assessment of social mindfulness in HRI to determine under which conditions, if any, we observe prosocial behavior towards robots. In Study One, participants interacted with a social robot (Cozmo) for three days, with perceived anthropomorphism and social mindfulness assessed before and after the interaction period. In Study Two, participants played the classic version of the SoMi paradigm using revised items matched in value for humans and robots, based on prior evaluations by a separate sample. Prolonged interaction with Cozmo did not increase social mindfulness but increased anthropomorphic perception of the robot. The revised items did not increase social mindfulness in the anthropomorphic condition, but they increased overall social mindfulness compared with previous studies. We conclude that real-life interaction does not necessarily enhance social mindfulness towards robots, that the item selection and their value for both human and robots must be considered, and that future studies should explore other interaction time frames and items.. Further, the increase in perceived anthropomorphism after a period of real-life interaction supports theory on anthropomorphism as a dynamic process. More general, the results stress that the field should carefully test HHI measures to ensure measurement validity before transferring them to HRI and that researchers must consider the context in which HRI occurs for external validity. Our findings also raise new questions for theory on social mindfulness, and support the emerging critique of the widely used

**Data availability statement:** The datasets/ Codes generated during and/or analysed during the current study are available in the osf repository, https://osf.io/xwvk2 and https://osf.io/8549p.

**Funding:** The author(s) received no specific funding for this work.

**Competing interests:** The authors declare that they have no known competing financial interests or personal relationships that could have appeared to influence the work reported in this paper.

Computers Are Social Actors (CASA) theory, which lead to the emergence of psychological measures in the field of HRI.

# 1. Introduction

## 1.1. Measuring social mindfulness in human robot interaction

It is expected that domestic robots will soon be part of our daily lives [1,2] and perform human tasks independently [3]. Especially with the prospect of social robots, one wonders what everyday interactions with these robots will look like. How will we treat them, and what design characteristics could influence interaction?

In response to such questions, the amount of research in the field of Human Robot Interaction (HRI) has grown exponentially [2,4–7]. While this rapidly growing body of research has yielded valuable insights, scholars argue that common terminology and research methods are lacking, leading to contradictory findings [5,8]. For example, it was pointed out that methodological choices made by researchers, such as stimuli or measures, directly affect research outcomes and that researchers should therefore reflect on methods, measures, and study design choices more meticulously [5,8]. Particularly in the field of pro-social behavior towards robots, as measured with the "Social Mindfulness" task, several methodological and ecological shortcomings of previous research limit the validity and reliability of its conclusions [7,9,10]. In the current research, we therefore aim to address these limitations with two studies and investigate whether and to what extent people show pro-social behavior towards robots in ecologically valid settings.

Before we elaborate on our research, we will summarize existing research and theories in this field.

## 1.2. Theoretical background

Extreme proponents of the Computers Are Social Actors (CASA) task or the Media Equation theory state that people interact with an object as if it was human if provided with enough social cues [11–14]. The most well studied social cue is anthropomorphism: the attribution of human features to non-human agents [15]. Research has shown that the more human-like a non-human agent appears to be in terms of its appearance or features, the more people accept it [16], empathize with it [17], and feel more interpersonal closeness to it [10]. For example, people trust autonomous cars more if they have anthropomorphic features [18]. However, it was also shown that increasing the anthropomorphic features of a robot does not always lead to better interaction or perception. For example, while human-likeness is correlated with empathy towards non-human agents, this only applies to a certain point of similarity; beyond that, our response reverts to aversion or even disgust; a phenomenon that has been termed the "uncanny valley" [19,20]. Many examples of the "uncanny valley" exist, with the android robot Sophia perhaps being the best example [21].

Apart from *perceptions* of non-human agents, the literature has also paid attention to *pro-social behaviors* towards non-human agents. For humans, pro-social behavior

is essential for successful interaction and cooperation [22,23] and affected by perceptions of trust [24], empathy [25], and interpersonal closeness [23,26]. Trust, empathy, and interpersonal closeness in HRI can also be affected by anthropomorphic features, posing the question how anthropomorphism might affect pro-social behavior towards robots. Increasing our knowledge about how human-like features and perception of robots can influence HRI is crucial to enable a smooth and safe interaction with robots in the future [27].

## 1.3. Social mindfulness

Building on the CASA theory, researchers propose that humans can even display pro-social acts towards non-human agents if enough social cues are provided [27]. This was tested by Wieringa and colleagues [28], who showed that if participants perceive robots to have a mind, they are willing to perform dull tasks to protect the robot from harassment, and Nijssen and colleagues [9], who found that with higher perceived anthropomorphism of robots, participants were more likely to sacrifice a group of *humans* to save a robot. Follow-up work investigated whether this behavior in extreme situations also translates to low-consequence, everyday situations; situations more akin to those participants could encounter with their domestic robot [29]. To test this question, Nijssen and colleagues transferred the Social Mindfulness task (SoMi) [30] from human-human interaction to HRI.

The SoMi task evaluates whether individuals consider the needs of others in social decision making, by letting participants choose from a set of mundane everyday items where not selecting unique items is considered the socially mindful choice, as it does not limit the other's choice (for a detailed description see methods section). Nijssen and colleagues (2021) did not find an effect of anthropomorphism on social mindfulness towards robots, while there was an effect of anthropomorphism on social mindfulness towards other humans. Whereas the results replicated the findings of Van Doesum and colleagues regarding HHI, they suggest that opposed to "life or death" situations [29], perceived anthropomorphism had no effect on every-day pro-social behavior towards robots.

These conclusions could have important implications for future design and implementations of domestic robots, suggesting that increased anthropomorphism does not cause humans to exhibit prosocial behaviour, and therefore potentially becoming emotionally dependent [31], to their domestic devices. Yet these are contradictory to findings showing that humans can and do create intense emotional and human-like relationships with non-human agents [32,33]. It is therefore important to know whether these conclusions are ecologically valid, and not caused by methodological choices [8].

Two main criticisms regarding the validity and methodology of previous research can be formulated. First and foremost, the results may be confounded by a lack of validity of the SoMi task in the context of HRI. As mentioned above, using psychological methods from the HHI literature (such as the SoMi task) to measure HRI is not new. In many such cases, the experimental design includes a baseline measure of some kind to ensure that the psychological measure is valid in the context of HRI [33,34] as it cannot be simply assumed that a measurement can be directly transferred from HHI to HRI. However, in the case of social mindfulness, previous research did not properly establish whether this is the case for the SoMi paradigm. Consequently, the null results regarding anthropomorphism and social mindfulness towards robots may have been caused by a lack of validity in the context of HRI, and not a truthful observation. Second, the results may not be valid simply because of certain ecological and methodological shortcomings that prevented the prosocial behavior to appear. More specifically, previous research with the SoMi task consisted of online studies, in which the robot was presented as a picture and the anthropomorphism manipulation was done with a vignette. Even though such procedures are common [28], especially in psychological research in HHI [35], it has also been criticized in the context of HRI [1]. In the current work, we argue that the two design features associated with these two criticisms (i.e., validity of SoMi task and lack of real-life experience with robots) may have had a major influence on the results. These two aspects will be investigated in two separate studies:

**1.3.1. Real-life interaction (Study 1).** Similar to the human version of the SoMi task [30], participants are presented with a picture of their robot partner, accompanied by a vignette. A previous study assessing the SoMi task only with

humans had assessed whether a SoMi effect also occurred when conducting the task in person [36]. As their results showed a significant SoMi effect regardless of methodology, Nijssen and colleagues assumed that this would count for conducting the task with robots as well.

However, in hindsight, this assumption is potentially problematic. Research has shown that the quality of HRI is likely to be biased after no real or only one time interaction [1]. This is most likely caused due to a lack of experience with a humanoid robot. This results in participants being unable to form expectations about said robots abilities, uses, or performance. Indeed, several studies have found differences in the ratings of the interaction after prolonged real-life interactions with a social robot compared to ratings after a short- or no interaction [37]. These entailed increased acceptance of the robot's shortcomings, more positive evaluations of the robot [1], increased meaningful social interaction [37], and settling into routines as participants familiarized with the robot [38].

Furthermore, in HHI, social mindfulness was more pronounced when the partner was physically present compared to a hypothetical interaction partner [36]. This poses the question whether physical presence of the robot interaction partner would induce asocially mindful response. Additionally, Nijssen and colleagues depict anthropomorphism as a static concept in their experimental set up, while recent evidence suggests that anthropomorphic perception of robots changes over time [39,40]. Hence, there is a need to study participants' pro-social behavior in association with perceived anthropomorphism, after a period of real-life interaction with a physical, social robot.

These shortcomings are addressed in Study 1 in which participants interact with a real, physical robot for three days. The participants conduct the SoMi task with the robot on the first and last day, to measure whether this real-life interaction effected the prosocial behavior of the participants towards the robot. The prolonged interaction with social robots to investigate the effects of real-life interaction on social mindfulness is new and increases the external validity of our experimental design. Further, this experimental setup allows for the investigation of changes in perceived anthropomorphism as a dynamic process. We thereby advance previous research and align our experimental design with the expected future implementation of domestic robots.

**1.3.2. Study stimuli (Study 2).** A key aspect of the SoMi task is that the items must be perceived as being equally relevant to both parties. For example, if you know that your partner is allergic to peanuts, you are more likely to choose the option containing peanuts regardless of whether this was the unique item or not. Therefore, if items are not seen as equally relevant to both parties, social mindfulness may still be occurring, just not in the way that is being measured.

Nijssen and colleagues transferred the SoMi task from HHI to HRI assuming that the items, whether unique or not, are relevant to both parties. As the items were designed to reflect humans' daily needs [30] one cannot blindly assume that the items of the SoMi task will be perceived as equally relevant to robots. Therefore, the participant's response might not reflect a lack of social mindfulness towards robots, but rather a perceived lack of item relevance for robots. For example, the SoMi task includes items such as a mug (robots don't consume), or a pen. In other words, the participant might think that none of the items are relevant for the robot, and hence they do not take the robot's preference into account as it seems unnecessary to do so. Therefore, a lack of a SoMi effect with the original items might not reflect a lack of prosocial behaviour towards robots but rather that the participant considers it impossible to understand the robots preference for any of these items at all.

In Study 2 we used participant evaluations to identify items that are equally relevant for both the human and robot interaction partners (as was the design for the HHI version). We then replicated the Nijssen (2021) study with these new items to determine whether this influenced prosocial behavior towards robots.

The overarching goal of this line of studies is to investigate whether and under which conditions humans might act pro-social towards domestic robots. Furthermore, we aim to investigate whether psychological measures like the SoMi task can be directly transferred to the human-robot context and how study design affects outcomes in HRI. Additionally, the revision of the items used in the SoMi paradigm can be seen as a first reflection on the human cognitive processes when taking other agents needs into account. It questions the fields assumptions about the mental processes involved in

 

prosocial behaviors and disentangles anthropomorphism (the attribution of human features to non-human agents) from the ascription of needs and values to non-human agents.

## 2. Study 1: Real-life interaction

Study 1 replicates and extends the Nijssen and colleagues (2021) study by having participants interact with a social robot (Cozmo) for three consecutive days in their home. Anthropomorphic perceptions of the robot as well as social mindfulness were assessed before and after the interaction period. The robot was previously recommended as an easy to use and low cost option for investigating real-life HRI [41].

This study therefore attempts to replicate the original HHI study that showed a SoMi effect also in real-life interactions, an assumption that Nijssen and colleagues' study was based on but had not been tested. We therefore hypothesize to see prosocial behaviour as measured by the SoMi paradigm after a real-life interaction compared to before. As previous studies have shown that extended interactions with a robot can influence the perception of said robot, participants will take the robot home with them for three days. This will maximize the chances of the participant forming some sort of expectation about what the robot does or does not prefer.

Furthermore, we predict that perceived anthropomorphism mediates the effect of social mindfulness in robots. Therefore, a lack of social mindfulness in general, as well as a lack of a mediation effect, would suggest that the SoMi task might not be able to measure prosocial behavior towards robots.

### 2.1. Methods

**Participants. Based on a power analysis using** G*Power [42] for a repeated measures ANOVA within-subjects design and medium effect size [43,44], the minimum required sample size for the current Study was $N = 28$ (based on $\alpha = .05$, $\beta = .80$, $f = .25$). The project was registered using the Open Science Framework and can be accessed via https://osf.io/xwvk2. Ethical approval was given by the Ethics Committee of Social Sciences, (Masked for peer review) (reference: ECSW-2019–006). In the current Study, $N = 32$ people participated (23 female) with a mean age of 21.16 years ($SD = 3.37$). Participant recruitment and data collection started at 01/03/2021 and ended at 30/10/2021. None of the participants had previously interacted with a social robot. All participants gave informed consent prior to participation.

**Materials.** Participants completed one questionnaire and one task. They completed this twice, once on Day 1 and again on Day 3 (see *Procedure*).

**Perceived anthropomorphism.** A subsection of the Godspeed questionnaire was used to measure perceived anthropomorphism [45]. This subsection consisted of 5 bipolar adjectives (e.g., fake vs. natural) on which Cozmo was rated, using a five-point Likert scale.

**Social mindfulness task.** The construct of social mindfulness was introduced and operationalized by Van Doesum and colleagues (2013). Social mindfulness differs from mindfulness by extending a predominantly self-oriented mindful awareness to include a benevolent perspective on the needs and wishes of others in the immediate social environment. The task is based on the idea that leaving or limiting choice is a subtle yet effective way to show benevolence, indifference, or even hostility towards others [46]. Van Doesum and colleagues conducted multiple studies showing that social mindfulness as measured by the SoMi task is defined by who the interaction partner is and how the relationship is characterized.

In this study, participants completed same the social mindfulness (SoMi) task used in the Nijssen and colleagues (2021) studies. Participants had to repeatedly choose among three items of the same category (e.g., pens). Crucially, two items are identical, while one item differs slightly in a certain aspect (e.g., two blue and one black pen). Specific examples of our materials can be found in the supplementary materials. Participants are told that they must choose an item, but that someone else (in this Study, the Cozmo robot) will pick something from the remaining items. It is counted how often the participant picks the socially mindful item, that is the item of which there are two, so the task partner still has a choice

between two unique items. The overall proportion of socially mindful versus non-socially mindful choices thus gives an indication of a participant's overall willingness to consider the task partner's needs.

The task consisted of two blocks: a solo condition and a social condition. The presentation of the blocks was counter-balanced, and each block consisted of 12 trials (6 test- and 6 distractor trials). During test trials, participants were presented with three items from which they could choose one. In all test trials, two objects were identical, and one object was different (e.g., two blue and one black pen). The distractor trials contained two of each type and was used to measure baseline preference. The presentation order of the test and distractor trials and the order of items were randomized.

In the solo condition, participants were instructed to imagine that they could take the chosen object home with them. In the social condition, participants were informed that someone else would choose between the remaining items.

**Cozmo.** Cozmo is a small robot originally designed by Anki (San Francisco, USA), now acquired by Digital Dream Labs (Pittsburgh, USA). One can interact with it through an app on either a smartphone or tablet. Cozmo can display emotions suited to the situation, recognize faces, and speak. Further, it can move freely in the room and play games with the cubes that come with it. By interacting with Cozmo, it learns, which leads to an increase of the functional use of the robot.

**Procedure.** After receiving Cozmo and an instruction letter at their house (see Appendix A), participants filled in the first survey on Qualtrics. The survey asked for demographic information (gender, age, previous robot interaction experience) and the Godspeed questionnaire for anthropomorphism. The survey ended with the social mindfulness task. After completion, the first day of the interaction period with Cozmo started (Day 1).

Participants were instructed to interact with Cozmo for at least 20 minutes each day for three consecutive days. We only limited the minimum interaction time per day, but participants were allowed to spend more time interacting with Cosmo with no upper interaction limit. We also controlled for the minimum interaction times with the Cosmo app. Participants could freely choose what to do with Cozmo, restricted only by Cozmo's functionalities. On the third day (Day 3), after their interaction time with Cozmo, participants had to fill in the second part of the survey. This asked the participants how long (in total) they interacted with Cozmo during the three days, the questionnaire for perceived anthropomorphism, and it ended with the social mindfulness task. Afterwards, participants could sign up for a lottery for a monetary reward and Cozmo would be picked up.

## 2.2. Analysis

**Questionnaire data.** A Principal Components Analysis was conducted to create the variable *Perceived Anthropomorphism (Anthr)* with a Cronbach's Alpha of 0.74. The factor scores were centered before being entered into the model.

**Linear mixed-effects models.** We used binomial linear mixed-effects models to analyze our data using the glmer function of the package lme4 [47] in R [48]. We coded socially mindful choices with 1, and 0 for non-socially mindful choices. The variable *Time* was sum-contrast coded (Day 1 versus Day 3) and only the test trials were included for social mindfulness (352 trials per time condition). The variable *Partner* was sum-contrast coded (Social vs Solo condition).

We aimed for a maximal random-effect structure, following the advice of [49]. This leads to the inclusion of random adjustment for repeated measures. Hence, a per-participant and per-item random adjustment was added to the fixed intercept, also known as random intercept. Lastly, all possible random correlation terms between the random effects were included.

## 2.3. Results

**2.3.1. No socially mindful responses.** The first model was a direct replication of the one reported in Nijssen et al. (2021), specifically the fixed effect *Partner*, with *Partner* as a random slope for both the participant and item random intercept. We used only data from Day 1 for this model, because the original study by Nijssen and colleagues only had

one measurement and participants in their study had not previously interacted with the robot. The model reported no significant difference between the Solo and the Social (Cozmo) condition in social mindfulness ($b=−0.07$, $SE=0.14$, $z=−0.53$, $p=.596$). Participants therefore did not change their choice performance based on whether someone/something would choose from the remaining items, replicating the Nijssen et al. (2021) results.

**No influence of interaction time/novelty.** To investigate whether the novelty effect played a role, a model was run with an interaction between *Partner* and *Time.* This model included all the data (Day 1 and Day 3) and included the *Partner\*Time* interaction as a random slope for both the participant and item random intercept. The model showed no significant effect of time on proportion of socially mindful responses ($b=0.07$, $p=0.497$; see Table 1). Figure 1 illustrates this result.

**2.3.2. Anthropomorphism influences social mindfulness over time.** There was a significant difference in perceived anthropomorphism on Day 1 ($M=−0.32$) compared to Day 3 ($M=0.32$, $t(34.45) = 2.30$, $p=.027$); replicating the novelty effect. We were interested in whether these ratings would influence social mindfulness with respect to Cozmo. This model only used data from the Social (Cozmo) condition. It contained an interaction term between *Time* and *Perceived Anthropomorphism (Anthr)* as both a fixed effect and a random slope for both the participant and item random intercept. The results are displayed in Table 2.

There is no significant effect of *Time* (Day 1 vs Day 3) or *Perceived Anthropomorphism* or the interaction of the two on socially mindful choices made in the Cozmo Condition. However, there is a tendency for the interaction effect, $p=.053$. Specifically, on Day 3, after the participants have had three days to interact with and get to know Cozmo, a higher perceived anthropomorphism results in *less* socially mindful choices. Figure 2 illustrates this tentative effect.

To further investigate the mediation effect of perceived anthropomorphism, we used the recommended Joint significance approach [50,51]. We tested the effect of time on social mindfulness in one model A (A path) and the effect of anthropomorphism on social mindfulness in another model B (B path). Model A did not yield a significant effect of time on social mindfulness ($b=0.22$, $SE=0.21$, $z=1.61$, $p=.107$, 95% CI[−0.05; 0.06]). Model B also indicated no significant effect of anthropomorphism on social mindfulness ($b=−0.22$, $SE=0.17$, $z=−1.24$, $p=.216$, 95% CI[−0.05; 0.06]). Therefore, we can exclude a mediation effect [52].

## 2.4. Discussion

Previous research by Nijssen and colleagues (2019; 2021) has shown that while participants display pro-social behavior towards robots in high-consequence situations (i.e., life or death), this does not replicate in low-consequence situations (i.e., choice between a yellow or blue pen). We argued in the introduction that before we accept this conclusion, we need to be sure that it is not an artefact caused by methodological shortcomings. It is possible that the original studies were not able to observe any prosocial behavior towards robots as the participant has no previous experience with the robot depicted in the vignette (as the robot is fictional) and hence could not form a prediction on what the robot does or does not prefer in terms of the SoMi items. To determine whether this is indeed the case, our Study 1 examined whether a

**Table 1. Binomial linear mixed model output for socially mindful choices.**

| Effect | Estimate | SE | z value | p |
|---|---|---|---|---|
| **Fixed effects** | | | | |
| Intercept | −0.64 | 0.17 | −3.75 | <.001*** |
| Partner | −0.14 | 0.11 | −1.23 | .218 |
| Time | 0.14 | 0.08 | 1.63 | .103 |
| Partner * Time | 0.07 | 0.10 | 0.68 | .497 |

*** <.001 N = 706, log-likelihood = -447.1

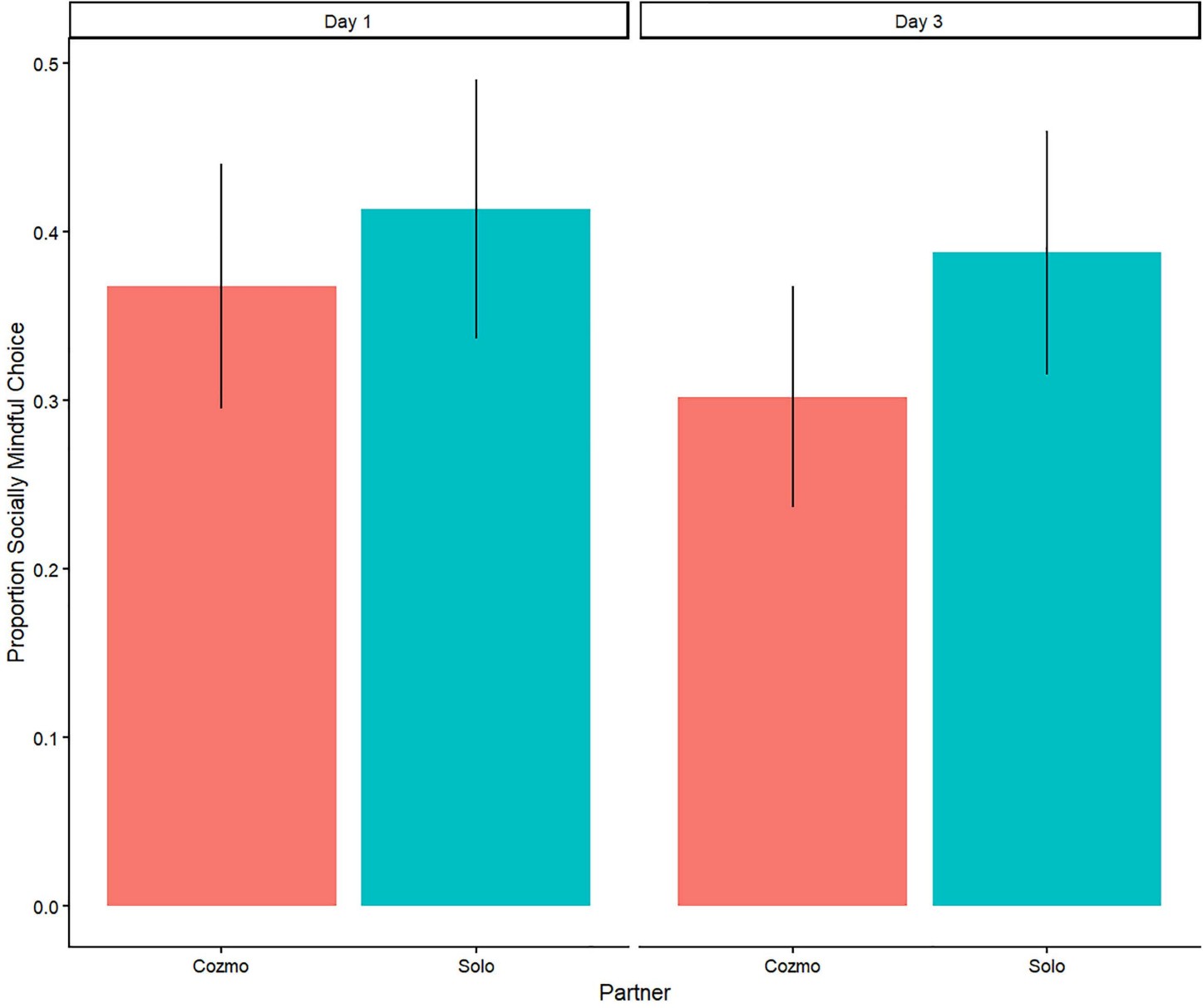

**Fig 1. Proportion of socially mindful responses on Day 1 and Day 3.** *Note.* The control condition (Solo) represents the preferences of the particiant. Therefore, the difference between the Solo and Cozmo condition represents the change in socially mindful choices elicited by the participant when they were told that Cozmo would choose from the items they do not pick. The figure shows no significant difference between the two conditions, regardless of whether the participant just "met" Cozmo (Day 1) or had been interacting with it for some time (Day 3; *p* = .497). Error bars represent 95% confidence intervals.

prolonged real-life interaction of 3 days with a social robot could cause prosocial behavior to emerge, as measured by the SoMi paradigm.

By using a physical robot instead of vignettes together with a prolonged interaction time, we aimed to resemble what domestic human robot interaction could look like, with the hypothesis that interactions during a longer period would

**Table 2. Binomial mixed model output for social mindful choices directed at Cozmo.**

| Effect | Estimate | SE | z value | p |
|---|---|---|---|---|
| **Fixed effects** | | | | |
| Intercept | −0.68 | 0.20 | −3.33 | <.001*** |
| Partner | −0.07 | 0.22 | −0.29 | .768 |
| Time | 0.20 | 0.17 | 1.13 | .257 |
| Partner * Time | 0.34 | 0.17 | 1.94 | .053 |

*** <.001 N = 366, log-likelihood = -221.2. *Note.* Anthr = Perceived Anthropomorphism.

increase socially mindful responses. However, the results revealed no prosocial behavior at any time point, similar to the original study [29].

Based on previous findings on the effect of anthropomorphism on perception of a mind behind the machine [10,17,29,53,54], we additionally investigated whether the effect of prolonged real-life interaction on social mindfulness was mediated by perceived anthropomorphism. Previous studies have shown a significant difference in the evaluations of a robot after a prolonged exposure compared to no exposure. We therefore hypothesized that we should also see a difference in perceived anthropomorphism over time.

Indeed, by comparing perceived anthropomorphism at Day 1 and Day 3, we were able to see a significant increase in anthropomorphism. Even though this contradicts previous HRI literature [39,40], it does show that the perception of the Cozmo robot significantly increases after a prolonged real-life interaction. This could suggest that participants indeed more successfully form expectation about the robot's capabilities.

However, there was no interaction or mediation effect of anthropomorphism and socially mindful behavior. Instead, we only found a tentative effect ($p = .053$) of perceived anthropomorphism on social mindful choices for Day 3 (post-interaction). Specifically, higher perceived anthropomorphism scores led to less socially mindful choices. This contradicts existing research on social cognition in HRI [1,29], which suggest instead that increased anthropomorphism leads to *increased* social mindfulness.

Our interim conclusion therefore is that using real-life interactions over a prolonged period of time does not cause pro-social behavior to emerge towards robots, as measured by the SoMi paradigm. As mentioned above, a lack of interaction with anthropomorphism in addition to this lack of prosocial behavior might also suggest that the SoMi task is not properly equipped to measure prosocial behavior towards robots. We turn to this potential confound in Study 2.

## 3. Study 2: Study stimuli

In Study 2, we explore the second potential shortcoming which relates to the items included in the SoMi task. In social interaction between humans, it has been shown that scarcity attracts: Without others involved, people tend to prefer the more unique item, regardless of cultural background [55]. Research has shown that leaving choice in the SoMi task is associated with other-orientedness, pro-social value orientations, empathic concerns, and perspective taking [36].

However, in human-human studies, it can be assumed that the items, whether unique or not, are relevant to both parties. As the items were designed to reflect humans' daily needs [30], one cannot safely assume that the items of the SoMi task will be perceived as equally relevant to robots. This is also illustrated by the example Van Doesum and colleagues give in their initial paper on the SoMi task (2013, p. 87) "*… being socially mindful can be as simple as not taking the last peanut butter cookie when there are still other alternatives left.*" What value would a peanut butter cookie have to a robot who is not able, does not need to, nor enjoys eating it? This rather methodological issue of measuring pro-social behavior towards non-human agents with a measurement tool that has been designed to represent human needs [30] has not yet been addressed and may have biased earlier research outcomes [29].

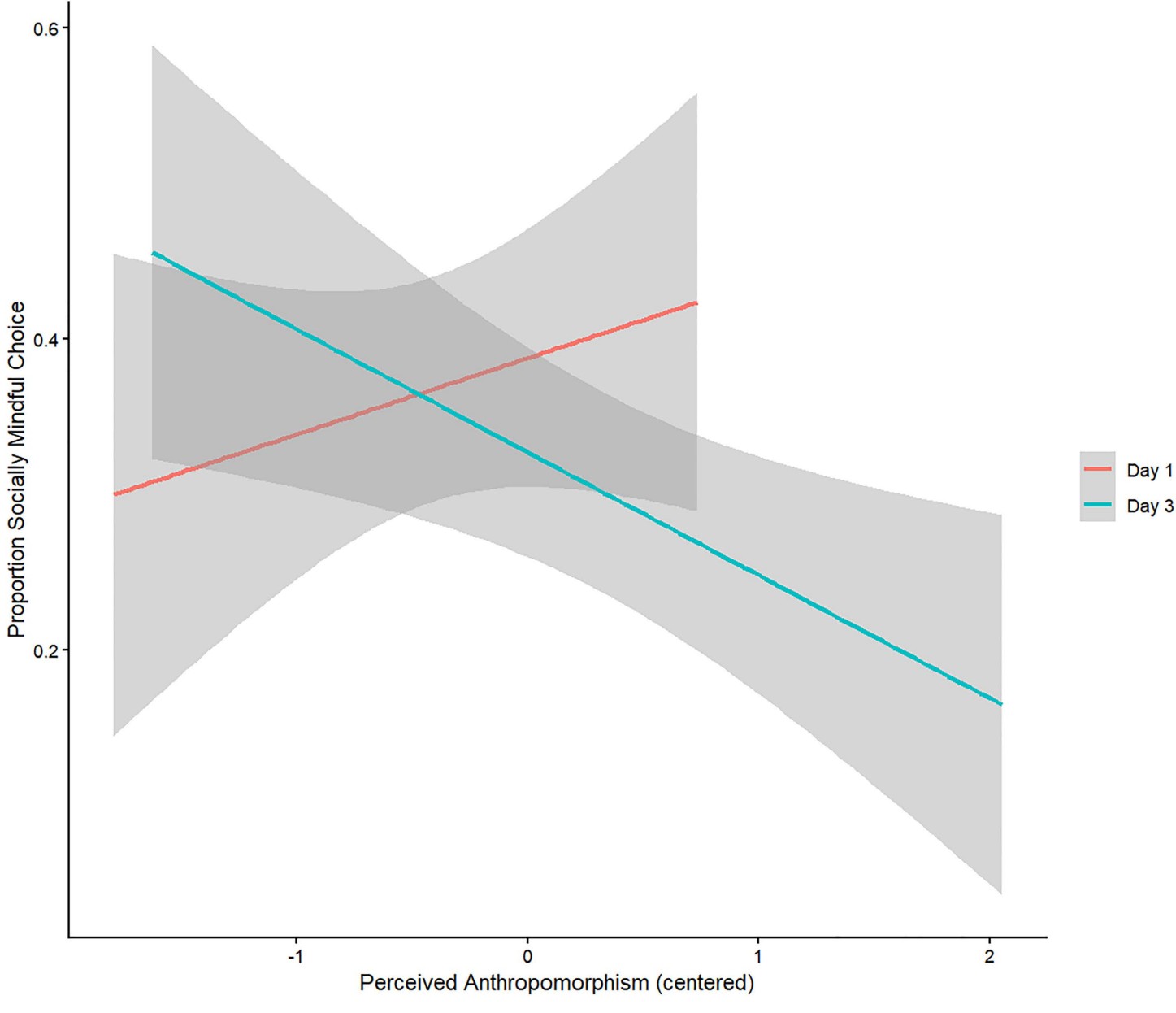

**Fig 2. Interaction between anthropomorphism, time, and social mindfulness.** *Note.* Error cloud represents standard error.

To determine whether the shortcoming regarding SoMi items could have influences the lack of prosocial behavior shown in earlier research, this study aim to create a stimulus set of items that are perceived as relevant to both humans and robots. Thereby, we aimed to create a SoMi task in which the participants would perceive the presented items as not only valuable to themselves or other humans (which is crucial to not make a social mindful choice too easy), but also as valuable for the robot (which is crucial to elicit social mindful behavior towards robots).

As the items need to matter for social mindful choices to be elicited [30], with this adjusted SoMi task we again aim to find prosocial behavior towards robots (H1) as well as the prosocial behavior being moderated by anthropomorphism (H2). These results would suggest that we can measure prosocial behavior towards robots using the SoMi task.

## 3.1. Methods

### 3.1.1. Object valuation task.
*Materials:* To come up with suitable items, we engaged a group of master students in a brainstorming activity which had the goal to find items that are valuable to both humans and robots. The students were following a research practicum, were familiar with the literature on HRI and the SoMi task. The final selection of items to be tested in an object evaluation task was then evaluated by the four researchers. This yielded 52 items (for example, a USB stick, antivirus software) that were included in the object valuation task. The items from the original SoMi task were also included, adding up to 61 items in total. The total list of items can be found in the supplementary materials.

*Participants:* A total of $N = 197$ persons agreed to participate in the object valuation task, which was accessed through the respective research institute's SONA system, linking to a Qualtrics Survey. The recruitment and participation started on 08/09/2025 and lasted until 10/10/2023. Of the participants, $N = 180$ ($M_{Age} = 19.84$, $SD = 2.02$; 146 females, 3 did not disclose) completed the entire questionnaire. Rewards in the form of SONA credits were provided. The experiment complied with the ethical guidelines of the research institution and receive ethical clearance (reference ECSW-LT-2023-2-16-77571).

*Procedure:* The participants were informed of the research's broad domain, without specific information on the scope and aims, along with the expected duration of 15 minutes. The participants were told that a succession of pictures, each presenting everyday objects, would be shown and that they were to rate the item in each picture as valuable to either humans or to robots. Order of the two blocks, humans or robots, was counterbalanced. They were asked to rate according to a scale that ranged from 0 to 10, with 0 being not at all valuable, and 10 being most valuable. Of note, the participants were instructed that "valuable" under the experiment's circumstances was to be understood not through a financial lens, but through one of *utility*. This was defined for the participants as "inexpensive objects can also be perceived as valuable since we are not looking for monetary value." The participants then rated a random selection of 22 items for human value (human block), and a random selection of 22 items for robot value (robot block). The participants' attention was checked via two check trials, one per block. Participants that did not pass the attention checks were *a priori* removed from the data set ($N = 0$).

*Item selection:* Each item's mean value score across participants was computed, with such a mean being obtained for humans and another for robots. There was no significant skew (skewness = (3 * (mean – median))/standard deviation) and thus, the mean was used. The human and robot mean for each item were then compared and the 12 items that had the smallest mean difference were selected for use in our conceptual replication of Nijssen and colleagues (2021). *T*-tests on these 12 items individually confirmed no significant difference in the ratings between the human and robot conditions ($M_{Robot} = 4.66$; $M_{Human} = 4.71$; *t* values ranged from 0 to 0.74; $p > .2$; see Table 3).

None of the 12 items contained the original SoMi items, as they were all rated as more relevant for the human ($M = 5.76$) than the robot ($M = 1.30$). This confirms part of our hypothesis claiming that the items of the original SoMi task are likely not perceived as having the same value for human and robot interaction partner. Whether adjusting for this with items having the same value was tested in our conceptual replication of Nijssen and colleagues (2021).

### 3.1.2. SoMi task.
*Participants:* We aimed for a sample of $N = 200$ participants, in order to replicate the conditions in the original Nijssen et al. (2021) Study. $N = 220$ participants started the online survey. Of these, participants were removed who did not complete the entire survey ($N = 17$) or their total time spent to complete the survey was 2.5 SDs longer than the average time ($M = 451.22s$, $SD = 558.62s$; $N = 3$). The final sample thus consisted of $N = 200$ participants ($M_{Age}$: 19.54 years, $SD_{Age}$: 1.89 years; 160 females, 1 undisclosed). The project was registered using the Open Science Framework and can be accessed via https://osf.io/8549p. The ethical approval for this study falls under the same approval for the "Masked for review" Study.

*Materials and procedure:* We used a copy of the original SoMi task as published in Nijssen et al. (2021) and only adjusted the items the participants could choose from. Within a single trial, the same items were presented in two different colors.

**Table 3. Valuation scores (out of 10) for each item scored for relevance for humans and robots.**

| Item | Human rating (mean) | Robot rating (mean) | Difference | *t* value |
|---|---|---|---|---|
| Radio Antenna | 4.36 | 5.44 | −1.08 | 0.02 |
| UV Foil | 2.68 | 3.37 | −0.69 | 0.09 |
| HDMI cable | 5.25 | 5.85 | −0.59 | 0.20 |
| Cable Organizer | 3.60 | 3.94 | −0.34 | 0.48 |
| USB key | 5.53 | 5.82 | −0.29 | 0.55 |
| Voltmeter | 4.64 | 4.48 | 0.15 | 0.74 |
| Cable pliers | 4.18 | 4.03 | 0.15 | 0.73 |
| Air Filter | 4.78 | 4.17 | 0.61 | 0.26 |
| Router | 6.89 | 6.00 | 0.89 | 0.08 |
| Air Duster | 3.77 | 2.82 | 0.96 | 0.03 |
| Microfiber cloth | 4.09 | 2.97 | 1.12 | 0.00 |

*Note.* These are the 12 items that were selected to be the stimuli in the online SoMi task for Study 2. In total 61 items were rated (once for humans, once for robots, counter balanced).

This experiment had a 2 (Partner: human vs. robot; within-subjects) by 2 (Condition: anthropomorphic vs. neutral; between-subjects) mixed design. Participants were randomly assigned to conditions. The materials and procedure were similar to Study 1, except that the Social (Cozmo) Condition was replaced by a vignette introducing a robot or a human. The presentation order of the human and robot block was counterbalanced. Each block consisted of an introduction to the task partner followed by 12 SoMi trials. Participants were introduced to their task partner with a vignette and a picture. In the anthropomorphic condition, these vignettes described the task partner in a humanized manner, that is, by emphasizing mental states such as their emotions and intentions. In the neutral condition, vignettes described the task partner in a neutral manner, without referring to their mental states (for more details about the vignettes, see [9,56,57]). A picture of the task partner was presented next to the vignettes. The combination of the vignettes and pictures was counterbalanced. The task partner was consistently referred to with a letter (e.g., "H"). Importantly, participants in the anthropomorphic condition read a humanized vignette for both the robotic partner and the human partner; while in the neutral condition, participants received a neutral vignette. The humanizing versus neutral manipulation effect of the vignettes was validated in previous work [9,57]. Subsequently, participants provided their age and gender, were thanked, debriefed, and awarded course credit.

**Data analysis:** We removed one baseline trial as it was coded incorrectly. We used the same data analysis procedure as described in Study 1. All conditions were sum-contrast coded.

### 3.2. Results

**Prosocial behavior towards robots.** To test whether our study elicited socially mindfulness responses, we compared the distractor and test trials. The distractor and test trials use the same stimuli, however, in the test trials, the participant could choose between three objects (two of one color and one in the other color), whereas for the distractor trials the participant can choose between four objects (two of one color, and two of the other color). The distractor trials therefore measure the participants color preference. If the participant chooses a different color (of the same item) in the test versus the distractor trials, we can assume that this is due to another influence than pure color preference.

We ran a binomial mixed model, with a *Partner* (Human vs Robot), *Trial Type* (Distractor vs Test), and *Condition* (Anthropomorphism vs Neutral) interaction as a fixed effect. *Partner* was a random slope for the participant and item random intercept. Table 4 reports the results.

**Table 4. Summary of the results of the binomial mixed effects model for the SoMi task with improved stimuli items.**

| Effect | Estimate | SE | z value | p |
|---|---|---|---|---|
| **Fixed effects** | | | | |
| Intercept | 1.21 | 0.17 | 7.20 | <.001*** |
| Partner | 0.10 | 0.05 | 2.00 | .048* |
| Condition | 0.11 | 0.08 | 1.44 | .149 |
| Trial Type | −0.25 | 0.04 | −6.61 | <.001*** |
| Partner * Condition | 0.00 | 0.05 | 0.04 | .970 |
| Partner * Trial Type | −0.03 | 0.04 | −0.75 | .456 |
| Condition * Type | −0.07 | 0.04 | −1.93 | .053 |
| Partner * Condition * Trial | −0.05 | 0.04 | −1.37 | .170 |

*** <.001 * <.05 N = 4748 log-likelihood = -2470.8

The results show a significant effect of *Trial Type* ($p < .001$) suggesting a significant difference in color choice depending on whether it was a test trial (1 unique item) or a distractor trial (2 of each item). This suggests that participants *did* show prosocial behavior with these new items. However, this is a summation of all the conditions (trial type as well as the presence of anthropomorphism).

To determine whether participants showed prosocial behavior towards their robot partner, we would expect to see *no* significant effect for the *Partner* by *Trial Type* interaction, which we do ($p = .456$). This suggests that the prosocial behavior does not differ between the partner types.

To determine whether the prosocial behavior was mediated by anthropomorphism, we would expect to see a significant effect for the *Trial Type* by *Condition* effect, which we do ($p = .053$). However, looking at the raw data, it is clear that this is mostly driven by the human partner. For this, there is an increase in prosocial behavior for the anthropomorphized human partner (11.3%) compared to the neutral human partner (2.6%). We do not see this for the robot partner (Anthropomorphized robot partner = 7%, neutral robot partner = 6.8%).

Therefore, even though we do see prosocial behavior with the relevant items, this effect is not moderated by anthropomorphism for the robot partner compared to the human partner. Additionally, when comparing these results to the Nijssen (2021) study, we see that the amount of prosocial behavior in general is a lot higher (Figure 3). For humans, it reaches 80%, which suggests that the participants are close to ceiling with these new items.

## 4. General discussion

The overarching aim of this research was to investigate whether and under which circumstances humans act pro-socially towards domestic robots. Previous work measuring prosocial behavior towards domestic robots have used the SoMi task, directly taken from human-human interaction studies [30]. However, two significant methodological criticisms could be formulated regarding prior studies' results: the lack of real-life interaction and the lack of task validity. The current research addressed these shortcomings in two respective studies, and aimed to determine *under which circumstances, if at all,* we could see prosocial behavior towards robots using the SoMi task.

In Study 1, we replaced the picture of a fictional robot with real-life interactions. From the human literature on the SoMi task, online versus physical versions of the task only resulted in a slightly more pronounced effect for the physical version. For this reason, Nijssen and colleagues chose the online version for their study. However, the robot literature states that a single sessions with an unknown, fictional robot, might influence how the participants perceive the robot's abilities (van der Graaf, 2015). Therefore, in Study 1, participants conducted the SoMi task with a real-life robot (Cozmo) and were even allowed to take it home with them for 3 days. As expected, perceptions of the robot significantly differed from the first

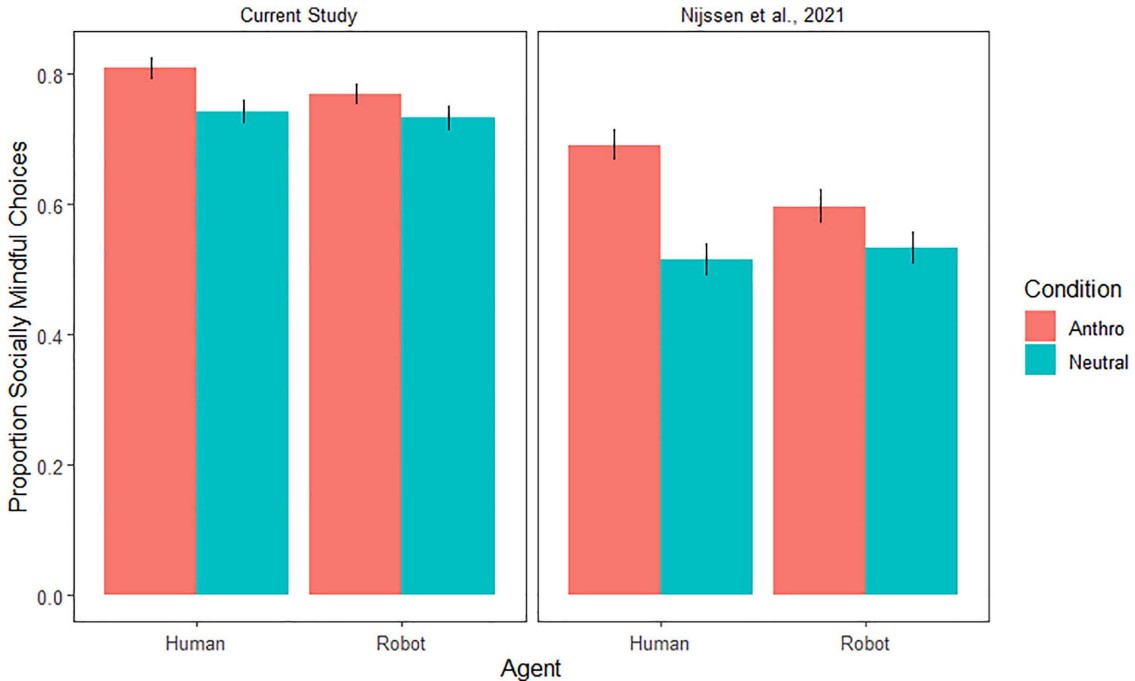

**Fig 3. Social mindfulness behavior the original Nijssen et al. (2021) study and the current study.** *This figure shows the prosocial behavior for the test trials only.*

impression (Day 1) compared to Day 3. Unfortunately, we did not see any effects on prosocial behavior towards the robot as measured by the SoMi task. Our findings on the change of anthropomorphism over time challenges the conception that anthropomorphism is a static state. Further, the difference in changes of perceived anthropomorphism and no changes in social mindfulness showed that prolonged interaction with social robots leads to faster changes in perceptions of robots than in behavior towards them. This also feeds into the discussion about the time frame, direction and theory on the novelty effect [1,37,38,58] and future research might elaborate on our findings by testing other interaction time frames tested with different social robots to understand under which circumstances we perceptual and behavioral changes towards robots occur.

For Study 2, we hypothesized that the original results may have been influenced by the presence of items that were not seen as relevant for the robot. This was confirmed in our evaluation of the original items and newly chosen ones: the items of the original SoMi task were rated as significantly more valuable for humans than for robots. Re-running the original Nijssen study with the improved items did result in prosocial behavior towards robots, albeit smaller than the human version from the Nijssen task. However, this prosocial behavior was not moderated by anthropomorphism. Participants exhibited almost the same prosocial behavior towards a neutral robot compared to an anthropomorphized robot. Unfortunately, anthropomorphism did influence the number of prosocial choices made with the human partner, but also here, less than the effect seen in the Nijssen task with the original stimuli. In that sense adjusting the items of the original SoMi task has consequences for social mindful choices of participants and this applies to both human-human and human-robot contexts. This is in line with others arguing that material, methods, and study design choices do influence research outcomes [5,8]. It also has implications for the original SoMi paradigm and theory on HRI.

In the original paper, the authors mention that the inspiration for the SoMi task came from these mini-jam-pots you typically find at a breakfast buffet [30]. Say you have two strawberry and one orange jam, and you know your friend will still

have to pick one after you, it is socially mindful to pick the strawberry one. But if you know that your friend does not like orange jam, or that they have an allergy to oranges, you can safely pick the orange one without being anti-social. Therefore, there are more elements at play than only the uniqueness of the items. This could also explain why changing the items resulted in changes in the prosocial behavior towards the human partner, and why anthropomorphism had a smaller influence on the strength of this behavior. For our SoMi trials with the new items, the effect of the items value on the social mindful decisions might go two ways. First, the new items that are now relevant to humans and robots, might be less valuable to humans than the original items of the SoMi task. Therefore, our research participants might experience less difficulty in letting the unique item go to another human or robot, as it is simply less valuable to themselves (It is easier to let a USB stick go than a good looking and expensive watch). In the robot trials this effect might be influenced by the additional perception that the new items also matter to the robot, so there is an incentive to not choose for the unique item and act socially mindful. This interaction between the items value for oneself, others and social mindful choices is something to explore deeper in future research on pro-sociality.

Further, we investigated whether psychological measures like the SoMi task can be directly transferred to the human-robot context. In general, neither longer interaction time and real-life interaction in Study 1, nor the relevance of the stimuli in Study 2 had an impact on SoMi results. This allows for two tentative conclusions: 1) we cannot conclude that the social mindfulness task can be blindly transferred to the HRI context andeven after altering the items and adapting the interaction context social mindfulness towards robots stays lower than towards humans. 2) pro-social behavior towards robots might *not* be exerted in the same manner as prosocial behavior towards humans.

The theoretical support in using human-human psychological tasks to measure HRI is based on the CASA theory which postulates that we interact with computers as if they are human, assuming they elicit social cues [9,11]. However, more and more evidence indicates that this assumption might not be true [59]. While CASA informs modern human machine application design, Heyselaar (2023) replicated the original study by Nass and colleagues (1994) and showed that participants no longer interact with computers as if they were humans. Further, studies using the Coin Entrustment game have found that people trust and cooperate differently with robots than other humans [60], and, using the prisoner dilemma task, it was found that humans tend to act more cooperatively towards fellow humans than robots [61]. In line with that, Heyselaar (2023) concluded that the CASA theory might only apply to emergent technology and that humans do not simply interact with technology as it was human. The current research feeds into the critique of the CASA paradigm, as transferring the SoMi task to robots did not make participants act prosocially towards robots in the same manner as they do to humans, even after adjusting the context and stimuli. Further, as indicated by robots application in elderly homes [62,63] and the development of domestic robots [1,8,38,64], robots might not be emergent technology anymore. Hence, HRI researchers should be careful in using older research findings and psychological measures transferred taken from HHI to build their research projects now.

These insights also have theory implication beyond the CASA paradigm. We have shown that without a reflection on the transferability of psychological measures, research might lead to misleading insights. Based on such insights, scholars form theories that in turn inform future research. Hence, we suggest caution and critical reflection on psychological measures transferability to the HRIO context to avoid misinformation feeding into theories.

## 5. Limitations and future research

While the current studies shed light into the application of the SoMi task to the human robot context and critically addressed confounds of previous studies, some limitations should be discussed. First, in Study 1, the duration of the interaction can be conceived of as a limitation. It is possible that any prosocial effects of long-term interactions with a social robot need more than three days to emerge [1,38,58]. Second, when replicating the study by Nijssen and colleagues (2021) with our revised items in Study 2, we aimed to stick to the original study set up as much as possible and hence, conducted the same online experiment. Future studies might therefore also test the adjusted version of the SoMi

in a real-life context or test the adjusted version after prolonged interaction with robots. While both an extended interaction period of more than three days and an application of the revised items to a prolonged interaction period would have been interesting to explore, the two studies were preregistered allowing only for minor changes and financial and temporal means did not allow for another study. Nevertheless, the changes in social mindfulness due to new items and the temporal changes in anthropomorphism found in the two studies highlight the potential of exploration in future research.

Moreover, the current research only employed quantitative methods which perhaps does not allow for the detection of more nuanced or layered psychological processes. In this respect, qualitative methods can yield interesting insight into the cognitive processes happening during HRI, which can possibly inform researchers on whether a psychological measurement tool can also be applied to HRI (c.f., Wieringa et al., 2023). Additionally, researchers might consider longitudinal and real-life study designs to increase the ecological validity of their findings. In line with previous works [5,8] we also advise scholars to critically reflect on the study design choices and materials they use, to avoid contradictive findings and generate fund outcomes. In general, we suggest that follow-up research investigates prosociality towards robots as independent of what we know about prosocial actions between humans.

## 6. Conclusion

The overarching aim of this research was to investigate whether and under which circumstances social mindfulness measures can be applied to HRI. Given the growing research in HRI and the field's problem when it comes to a lack of common terminology and research methods, we focused on of pro-social behavior towards robots, as measured with the "Social Mindfulness" task and addressed several shortcomings of previous work. The outcomes illustrated the difficulties and dangers that researchers face when applying psychological measures from HHI to the HRI context. Our findings highlight that researchers must carefully reflect on the validity of their measures and external validity of their research design to avoid confounded conclusions that feed into theories on HRI. With this line of work, we hope to inspire future research focusing on the adaptation and improvement of measurement instruments to the HRI literature.

## Author contributions

**Conceptualization:** Dennis Nientimp, Barbara C. N. Müller, Evelien Heijselaar.

**Data curation:** Dennis Nientimp.

**Formal analysis:** Dennis Nientimp, Evelien Heijselaar.

**Methodology:** Dennis Nientimp, Barbara C. N. Müller, Sari R. R. Nijssen, Evelien Heijselaar.

**Project administration:** Dennis Nientimp, Barbara C. N. Müller.

**Supervision:** Barbara C. N. Müller, Sari R. R. Nijssen, Evelien Heijselaar.

**Visualization:** Dennis Nientimp, Evelien Heijselaar.

**Writing – original draft:** Dennis Nientimp, Barbara C. N. Müller, Evelien Heijselaar.

**Writing – review & editing:** Dennis Nientimp, Barbara C. N. Müller, Sari R. R. Nijssen, Evelien Heijselaar.

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
