## [Decision Letter · Decision Letter 0]

20 Oct 2025

PONE-D-25-47098One size fits all? Transferring Social Mindfulness Measures to HRI.PLOS ONE?

Dear Dr. Nientimp,

Thank you for submitting your manuscript to PLOS ONE. After careful consideration, we feel that it has merit but does not fully meet PLOS ONE’s publication criteria as it currently stands. Therefore, we invite you to submit a revised version of the manuscript that addresses the points raised during the review process.

If applicable, we recommend that you deposit your laboratory protocols in protocols.io to enhance the reproducibility of your results. Protocols.io assigns your protocol its own identifier (DOI) so that it can be cited independently in the future. For instructions see: https://journals.plos.org/plosone/s/submission-guidelines#loc-laboratory-protocols . Additionally, PLOS ONE offers an option for publishing peer-reviewed Lab Protocol articles, which describe protocols hosted on protocols.io. Read more information on sharing protocols at https://plos.org/protocols?utm_medium=editorial-email&utm_source=authorletters&utm_campaign=protocols.

We look forward to receiving your revised manuscript.

Kind regards,

Zeashan Hameed Khan, Ph.D.

Academic Editor

PLOS ONE

Journal Requirements:

2. Please remove your figures from within your manuscript file, leaving only the individual TIFF/EPS image files, uploaded separately. These will be automatically included in the reviewers’ PDF.

Additional Editor Comments:

The paper has been evaluated, however, some major improvements are needed to enhance the quality of this research. Please re-submit the improved version with a point by point explanation of the reviewer's comments.

Reviewers' comments:

Reviewer's Responses to Questions

**Comments to the Author**

1. Is the manuscript technically sound, and do the data support the conclusions?

Reviewer #1: Partly

Reviewer #2: Yes

2. Has the statistical analysis been performed appropriately and rigorously?

Reviewer #1: Yes

Reviewer #2: Yes

3. Have the authors made all data underlying the findings in their manuscript fully available?

Reviewer #1: Yes

Reviewer #2: Yes

4. Is the manuscript presented in an intelligible fashion and written in standard English?

Reviewer #1: Yes

Reviewer #2: Yes

Reviewer #1: 1. The abstract effectively identifies a methodological gap but provides little detail on how the two studies were structured or what types of robots and interaction settings were used. Clarifying the experimental design and the scope of interaction contexts would help readers better understand the validity and generalizability of the findings.

2. While the paper challenges the direct transfer of HHI measures to HRI, the abstract lacks a clear articulation of what new insights or frameworks emerge from this work. The authors should elaborate on how their findings advance the theoretical understanding of social mindfulness in HRI or guide future methodological development in robot interaction research.

3. The introduction should clearly conclude with a distinct paragraph that highlights the novel contributions of your work.

4. The literature review should benefit from more explorations of previous studies.

5. The discussion section needs to be expanded to more thoroughly analyze the results.

6. The first paragraph of the conclusion should succinctly summarize the contributions of the study in past tense.

7. The second paragraph of the conclusion should provide clear and actionable future recommendations.

8. Some equations are not properly cited.

9. Sections and subsections are not numbered and this confuses the division of the manuscript.

Reviewer #2: This article provides a clear and critical assessment of how the Social Mindfulness Paradigm (SoMi), originally developed to measure prosocial behavior between humans, applies to human-robot interaction (HRI). The key finding is that the traditional SoMi task may not validly capture prosociality in HRI. The authors highlight the need for methodological adjustments when transferring human-based psychological tools to robots, especially regarding stimulus relevance, ecological validity, and anthropomorphism.

The study addresses an interesting and emerging research area and presents a thorough investigation. However, a few questions arise:

1. If the authors recognized that a three-day interaction might be too short for prosocial behavior to develop, why wasn’t the duration extended?

2. In Study 1, what was the rationale for limiting daily interaction with Cozmo to at least 20 minutes rather than allowing longer sessions?

**Do you want your identity to be public for this peer review?** For information about this choice, including consent withdrawal, please see our Privacy Policy

Reviewer #1: No

Reviewer #2: No

---

## [Author Response · Author response to Decision Letter 1]

17 Nov 2025

Dear Reviewers,

Thank you both for your comments and for taking the time and effort to provide such in depth feedback. Your comments are insightful and stimulated an academic discussion among the research team that in our opinion further improved the paper. Below we respond to each of the points you raised separately and explain how and where we addressed them in the revised version of the manuscript (the revised version entails the track changes and is also the version that the line numbers refer to):

Reviewer 1:

Reviewer #1: 1. The abstract effectively identifies a methodological gap but provides little detail on how the two studies were structured or what types of robots and interaction settings were used. Clarifying the experimental design and the scope of interaction contexts would help readers better understand the validity and generalizability of the findings.

- We added the requested information and agree that providing more information on robots used, experimental setup and interaction context in the abstract makes it easier for others to understand the meaning, context and generalizability of our research.

2. While the paper challenges the direct transfer of HHI measures to HRI, the abstract lacks a clear articulation of what new insights or frameworks emerge from this work. The authors should elaborate on how their findings advance the theoretical understanding of social mindfulness in HRI or guide future methodological development in robot interaction research.

- We added a few sentences on theory on social mindfulness in robots and anthropomorphism as a dynamic perceptual concept. While our findings have theory implication, we believe that the main contribution of our research is methodological. Our outcomes show how sensitive research measures are to context and experimental design. Hence, we also highlighted that future research should engage in cautious reflection when transferring measures from HHI to HRI, in our abstract.

- Based on your remark we also added a few sentences about theory contributions and methodological development to the discussion/future research section of the paper.

3. The introduction should clearly conclude with a distinct paragraph that highlights the novel contributions of your work.

- Below the introduction of both study one and study two (line 194 and 227) we added two paragraphs that highlight the novel contribution of each respective study to the field.

4. The literature review should benefit from more explorations of previous studies.

- We have included references throughout the text, merging results from HHI and HRI in terms of SoMi to highlight better the research gap our manuscript aims to elucidate. We also want to stress that we used Vancouver referencing style as requested by PlosOne, which means references are only included as numbers in the manuscript and might therefore easily be overlooked. We also reformulated our discussion of earlier research in some parts to elaborate more on previous research findings (e.g. line 164 or 430 onwards). However, in the case that you know of additional relevant literature we would be thankful for any recommendation.

5. The discussion section needs to be expanded to more thoroughly analyze the results.

- Thank you for this remark. We agree that a more thorough interpretation of the results in the discussion section would also make the connection to theory, methods and previous research more obvious for the reader. We added multiple sentences on the interpretation of our result to the discussion (see lines: 640-648,672-681, 687)

6. The first paragraph of the conclusion should succinctly summarize the contributions of the study in past tense.

- The manuscript which you reviewed had no conclusion paragraph, but we added one at the end of the manuscript and stuck to the past tense when it was grammatically right to do so.

7. The second paragraph of the conclusion should provide clear and actionable future recommendations.

- We added concrete recommendations for future research to multiple sections in our discussion (see lines: 644, 680, 706-713, 728, 749)

8. Some equations are not properly cited.

- We looked through the manuscript and could figure out which equations you were referring to. Would you be so kind and tells us the line in which you found them, so that we can adjust the missing reference?

9. Sections and subsections are not numbered, and this confuses the division of the manuscript.

- Thank you for pointing this out, we added numbers to all section to make the division of the manuscript clearer.

Kind regards,

The research team

Reviewer 2

Reviewer #2: This article provides a clear and critical assessment of how the Social Mindfulness Paradigm (SoMi), originally developed to measure prosocial behavior between humans, applies to human-robot interaction (HRI). The key finding is that the traditional SoMi task may not validly capture prosociality in HRI. The authors highlight the need for methodological adjustments when transferring human-based psychological tools to robots, especially regarding stimulus relevance, ecological validity, and anthropomorphism.

The study addresses an interesting and emerging research area and presents a thorough investigation. However, a few questions arise:

1. If the authors recognized that a three-day interaction might be too short for prosocial behavior to develop, why wasn’t the duration extended?

Dear Reviewer 2, there are two reasons for not extending the duration:

- Reason 1: Preregistration and research gap: At the time when study 1 was conducted, there was not much literature on long time interactions with robots and their effect on pro-social behavior. Therefore, we designed and pre-registered the experiment with the belief that three-days were enough to have a meaningful experimental manipulation. When looking at measures regarding perceived anthropomorphism we also see that the manipulation worked. We have additional measures on perceived warmth and competence also indicating chages, which we did not include in the paper as they are outside the scope of the current manuscript. Further, at the time we did not yet know the outcomes of our manipulation as we pre-registered the experiment so that the data would be analyzed after they were collected, to not confound our data collection by trying to confirm our hypothesis.

- Reason 2: Time and money restrictions. The data was collected by 3 research master students one of whom is author on this paper. Their study trajectory did not allow for another time window. Further, the money to compensate the participants was not enough to ask for more time investments from the participants' side.

However, we agree with you that an extended period of interaction would have been interesting to explore and in hindsight we would have wished for more funding and the means to conduct such research. Hence, added this as something for future studies to explore and explained our constraints (line 727-732). Still, we believe that study 1 was insightful. It showed that in contrast to our expectations social mindfulness as a behavioral measure (at least with the set of items included in study 1) does not change as fast as perceptual measures like anthropomorphism. This is something to build on in future research that wants to investigate changes over time.

2. In Study 1, what was the rationale for limiting daily interaction with Cozmo to at least 20 minutes rather than allowing longer sessions?

- We think there is a misunderstanding of our wording here. This is a lower limit we write about. It means the participants had to interact with Cozmo for a minimum of 20 minutes. We see that reader might get confused and therefore added a more precise description to the manuscript (line 324-326):

“Participants were instructed to interact with Cozmo for at least 20 minutes each day for three consecutive days. We only limited the minimum interaction time per day, but participants were allowed to spend more time interacting with Cosmo with no upper interaction limit. We also controlled for the minimum interaction times with the Cosmo app.”

Kind regards,

The research team

---

## [Decision Letter · Decision Letter 1]

24 Nov 2025

PONE-D-25-47098R1One size fits all? Transferring social mindfulness measures to HRI.PLOS ONE?

Dear Dr. Nientimp,

If applicable, we recommend that you deposit your laboratory protocols in protocols.io to enhance the reproducibility of your results. Protocols.io assigns your protocol its own identifier (DOI) so that it can be cited independently in the future. For instructions see: https://journals.plos.org/plosone/s/submission-guidelines#loc-laboratory-protocols . Additionally, PLOS ONE offers an option for publishing peer-reviewed Lab Protocol articles, which describe protocols hosted on protocols.io. Read more information on sharing protocols at https://plos.org/protocols?utm_medium=editorial-email&utm_source=authorletters&utm_campaign=protocols.

We look forward to receiving your revised manuscript.

Kind regards,

Zeashan Hameed Khan, Ph.D.

Academic Editor

PLOS ONE

Journal Requirements:

Additional Editor Comments (if provided):

The paper has significantly improved. However some minor corrections are required to further polish the results.

Some recent reviews in collaboration and HRI may also help if relevant, such as: https://doi.org/10.1177/18761364251393221.

Reviewers' comments:

Reviewer's Responses to Questions

**Comments to the Author**

Reviewer #1: (No Response)

Reviewer #2: All comments have been addressed

2. Is the manuscript technically sound, and do the data support the conclusions?

Reviewer #1: (No Response)

Reviewer #2: Yes

3. Has the statistical analysis been performed appropriately and rigorously?

Reviewer #1: (No Response)

Reviewer #2: Yes

4. Have the authors made all data underlying the findings in their manuscript fully available?

Reviewer #1: (No Response)

Reviewer #2: Yes

5. Is the manuscript presented in an intelligible fashion and written in standard English?

Reviewer #1: (No Response)

Reviewer #2: Yes

Reviewer #1: Well responded, however, the abstract still lack more info from the results, it is overly lengthy on general information.

Reviewer #2: The authors have addressed all of my querries. I recommend the publication of this article to PLOS ONE Journal.

**Do you want your identity to be public for this peer review?** For information about this choice, including consent withdrawal, please see our Privacy Policy

Reviewer #1: No

Reviewer #2: No

---

## [Author Response · Author response to Decision Letter 2]

4 Dec 2025

Dear Reviewers,

Thank you for reading our manuscript again and providing your feedback. Your feedback has improved the manuscript, and we want to thank you for that. We are very happy that you deem many of your comments addressed adequately in our revised version of the manuscript. Let us react to the remaining ones.

Comments and reactions:

Reviewer #1: Well responded, however, the abstract still lack more info from the results, it is overly lengthy on general information.

Dear Reviewer 1,

We tried to address your previous concerns regarding the abstract in our last iteration of the manuscript but from your comment we deduce that you are not yet satisfied. We repeat your previous comments regarding the manuscript below for clarity.

1. The abstract effectively identifies a methodological gap but provides little detail on how the two studies were structured or what types of robots and interaction settings were used. Clarifying the experimental design and the scope of interaction contexts would help readers better understand the validity and generalizability of the findings.

2. While the paper challenges the direct transfer of HHI measures to HRI, the abstract lacks a clear articulation of what new insights or frameworks emerge from this work. The authors should elaborate on how their findings advance the theoretical understanding of social mindfulness in HRI or guide future methodological development in robot interaction research.

As stated by you we are very happy to hear we have responded your comments. We added information on how the studies were structured, the types of robots used and the experimental design, to allow for judgements regarding generalizability:

“Addressing these methodological shortcomings, the current studies investigated whether the SoMi task is a valid assessment of social mindfulness in HRI to determine under which conditions, if any, we can measure prosocial behavior towards robots. In Study One, participants interacted with a social robot (Cozmo) for three days, with perceived anthropomorphism and social mindfulness assessed before and after the interaction period. In Study Two, participants played the classic version of the SoMi paradigm using revised items matched in value for humans and robots, based on prior evaluations by a separate sample. Prolonged interaction with Cozmo did not increase social mindfulness but increased anthropomorphic perception of the robot. The revised items did not increase social mindfulness in the anthropomorphic condition, but they increased overall social mindfulness compared with previous studies.”

And we addressed validity and advances for theory and methodology in our abstract:

“Based on mixed findings, we stress that the field should carefully test HHI measures to ensure measurement validity before transferring them to HRI and that researchers must consider the context in which HRI occurs for external validity. Further, our findings contribute to theory on anthropomorphism as a dynamic process, raise new questions for theory on social mindfulness, and support the emerging critique of the widely used CASA paradigm.”

We believe that you were rightfully asking for the addition of these points and very much agree that they improved the abstract.

To react on your most recent comment we revised the abstract once more and tried to find a balance between what you asked us to add during the first review and the current comment.

1. “lack more info from the results “

We added some more information regarding the results and how they tie in with existing literature, future work and theory (additions marked by brackets):

“Prolonged interaction with Cozmo did not increase social mindfulness but increased anthropomorphic perception of the robot. The revised items did not increase social mindfulness in the anthropomorphic condition, but they increased overall social mindfulness compared with previous studies. [We conclude that real-life interaction does not necessarily enhance social mindfulness towards robots, that the item selection and their value for both human and robots must be considered, and that future studies should explore other interaction time frames and items. Further, the increase in perceived anthropomorphism after a period of real-life interaction supports theory on anthropomorphism as a dynamic process.] More general, the results stress that the field should carefully test HHI measures to ensure measurement validity before transferring them to HRI and that researchers must consider the context in which HRI occurs for external validity.”

These are our main findings, and they answer 1) the question whether prolonged interaction with social robots increases social mindfulness and how it affects anthropomorphic perception of it, and 2) whether item value matters for the outcomes of the social mindfulness paradigm. This answers our main research questions. In case this was not the addition you were hoping for please point out specifically what you are missing.

2. “it is overly lengthy on general information”

While we are well within the limits set by PlosOne publishing guidelines the length of the abstract indeed increased. Hence, we tried to carefully shorten the abstract and remove redundant and overly general information. We say “carefully” as from your previous comment we understood that you missed some information on “detail on how the two studies were structured or what types of robots and interaction settings were used” and experimental design and the scope of interaction contexts” to enhance understandability of validity and generalizability for readers. We think you raised an excellent point there and wanted to keep this improvement. Nevertheless, we managed to cut the abstract from 399 to 341 words (see document for the full abstract). We hope that the current adjustments do justice on both your previous comments and your new comments.

Reviewer #2: The authors have addressed all of my querries. I recommend the publication of this article to PLOS ONE Journal.

Dear Reviewer 2,

Thank you for approving our manuscript. Through your comments were able to increase the quality of it and we are very thankful for that.

Kind regards,

The research team

---

## [Decision Letter · Decision Letter 2]

16 Dec 2025

One size fits all? Transferring social mindfulness measures to HRI.

PONE-D-25-47098R2

Dear Dr. Nientimp,

We’re pleased to inform you that your manuscript has been judged scientifically suitable for publication and will be formally accepted for publication once it meets all outstanding technical requirements.

Kind regards,

Zeashan Hameed Khan, Ph.D.

Academic Editor

PLOS One

Additional Editor Comments (optional):

The revised version has significantly improved in quality and content. Therefore, it can be accepted in the present form.

Reviewers' comments:

Reviewer's Responses to Questions

**Comments to the Author**

Reviewer #1: (No Response)

Reviewer #2: All comments have been addressed

2. Is the manuscript technically sound, and do the data support the conclusions?

Reviewer #1: (No Response)

Reviewer #2: (No Response)

3. Has the statistical analysis been performed appropriately and rigorously?

Reviewer #1: (No Response)

Reviewer #2: (No Response)

4. Have the authors made all data underlying the findings in their manuscript fully available?

Reviewer #1: (No Response)

Reviewer #2: (No Response)

5. Is the manuscript presented in an intelligible fashion and written in standard English?

Reviewer #1: (No Response)

Reviewer #2: (No Response)

Reviewer #1: (No Response)

Reviewer #2: (No Response)

**Do you want your identity to be public for this peer review?** For information about this choice, including consent withdrawal, please see our Privacy Policy

Reviewer #1: No

Reviewer #2: No

---

## [Editor Report · Acceptance letter]

PONE-D-25-47098R2

PLOS One

Dear Dr. Nientimp,

I'm pleased to inform you that your manuscript has been deemed suitable for publication in PLOS One. Congratulations! Your manuscript is now being handed over to our production team.

Kind regards,

on behalf of

Dr. Zeashan Hameed Khan

Academic Editor

PLOS One